# African Swine Fever in Two Large Commercial Pig Farms in LATVIA—Estimation of the High Risk Period and Virus Spread within the Farm

**DOI:** 10.3390/vetsci7030105

**Published:** 2020-08-07

**Authors:** Kristīne Lamberga, Edvīns Oļševskis, Mārtiņš Seržants, Aivars Bērziņš, Arvo Viltrop, Klaus Depner

**Affiliations:** 1Veterinary Surveillance Department, Food and Veterinary Service, Peldu iela 30, LV-1050 Riga, Latvia; Edvins.Olsevskis@pvd.gov.lv (E.O.); martins.serzants@pvd.gov.lv (M.S.); 2Faculty of Veterinary Medicine, Latvia University of Life Sciences and Technologies, Liela iela 2, LV-3001 Jelgava, Latvia; aivars.berzins@bior.lv; 3Institute of Food Safety, Animal Health and Environment BIOR, Lejupes Street 3, LV-1076 Riga, Latvia; 4Institute of Veterinary Medicine and Animal Sciences, Estonian University of Life Sciences, Fr.R.Kreutzwaldi 1, 51006 Tartu, Estonia; arvo.viltrop@emu.ee; 5Friedrich-Loeffler-Institut (FLI), Federal Research Institute for Animal Health, Südufer 10, 17493 Greifswald-Insel Riems, Germany; Klaus.Depner@fli.de

**Keywords:** African swine fever, commercial farms, early detection, high risk period, disease control

## Abstract

African swine fever (ASF) was first detected in Latvia in wild boar at the Eastern border in June 2014. Since then ASF has continued to spread in wild boar populations covering almost whole territory of the country. Sporadic outbreaks occurred at the same time in domestic pig holdings located in wild boar infected areas. Here we present the results of the epidemiological investigation in two large commercial farms. Several parameters were analyzed to determine the high risk period (HRP) and to investigate the ASF virus spread within the farm. Clinical data, mortality rates and laboratory results proved to be good indicators for estimating the HRP. The measures for early disease detection, particularly the enhanced passive surveillance that is targeting dead and sick pigs, were analyzed and discussed. Enhanced passive surveillance proved to be a key element to detect ASF at an early stage. The study also showed that ASF virus might spread slowly within a large farm depending mainly on direct contacts between pigs and the level of internal biosecurity. Findings suggest improvements in outbreak prevention, control measures and may contribute to a better understanding of ASF spreading patterns within large pig herds. Culling of all pigs in large commercial farms could be reconsidered under certain conditions.

## 1. Introduction

The first set of cases of African swine fever (ASF) in Latvia were detected in the wild boar population at the eastern border with Belarus in June 2014 [1,2,3]. The disease continued to spread over most of the territory of Latvia within the following years [4,5,6]. Similar to other ASF affected countries, the infection cycle is maintained by wild boar and this remains a permanent threat for the domestic pig industry [1,7,8]. In parallel, to the epidemic in wild boar, sporadic ASF outbreaks occurred mostly in backyard farms located in areas where wild boar were affected [1]. Due to their low level of biosecurity, backyard farms are more vulnerable to ASF virus introduction, while large commercial pig farms are better protected although biosecurity may vary from farm to farm [9]. The first outbreak in a large commercial farm occurred in January, 2017 followed by a secondary outbreak three weeks later in another farm owned by the same company [10].

In total 37.744 domestic pigs died or had to be culled so far due to ASF. An overview of the ASF epidemic in Latvia since 2014 is shown in Table 1.

In case of an ASF outbreak, a timely epidemiological investigation has to be conducted to identify the possible source and to estimate the high risk period (HRP), which is the likely length of time that ASF has been present on the farm. It is very important to detect infected farms as early as possible after ASF virus introduction to avoid further spread and to minimize the losses to the pig sector and reduce governmental costs associated with outbreak eradication [14]. Animal experiments as well as field observations have shown that a low to moderate transmissibility of ASF virus between pigs is leading to a slow spread of the disease within a farm [10,15,16,17,18,19]. Therefore, the inadvertent introduction of the virus may remain unnoticed by the farmer or the veterinarian for extended time. Under unfavorable circumstances, the HRP may be several weeks or even months during which the virus can spread within a farm or to other farms.

The main strategic aim of ASF surveillance in domestic pigs is to keep the HRP as short as possible by early detection of infected holdings. To facilitate the early detection of outbreaks, new measures were introduced in Latvia in 2017 requiring regular sampling and testing of sick and dead pigs for ASF virus. Each week, at least the first two deaths, including post weaning pigs or pigs older than two months in each production unit, had to be sampled and tested [12,20]. This enhanced passive surveillance approach was based on the assumption that due to the high case fatality (>90%) of ASF almost all infected pigs will become sick and die [15,21,22]. Therefore, any sick or dead animal would be a good candidate for ASF testing.

In this report, we present the results of epidemiological investigations conducted in two large pig farms, affected by ASF in Latvia. One outbreak (Farm A) occurred in 2017 before the enhanced passive surveillance was introduced, while the second outbreak (Farm B) occurred in 2018, when enhanced surveillance was already in place. We focused on the early detection strategy, the HRP and the virus spread within the farms. Our case report contributes to a better understanding of ASF spread patterns within large pig herds and suggests how outbreak prevention and control measures can be improved.

## 2. Case Description

### 2.1. Farm A

Farm A is located in south-east of Latvia, a farrow to finish farm with almost 6000 pigs, including nearly 600 breeding sows notified ASF suspicion to the veterinary authority on 14th July 2017 and the outbreak was confirmed by the veterinary authority at the same day. At that time, an active spread of the infection was ongoing in the local wild boar population [23] (Figure 1). The closest ASF positive wild boar was found less than 3 km from the farm. Pigs were kept indoors in seven separate stables according to their production category (sows, boars, gilts and finishers). Sows were kept in three stables depending on their reproduction phase—sows for artificial insemination, pregnant sows kept in groups and farrowing sows. The farm had its own slaughterhouse located within the farm’s territory. The meat from the slaughterhouse was delivered to meat processing and retail establishments in Latvia.

### 2.2. Farm B

Farm B, a large farrow to finish farm with around 16.000 pigs, including nearly 1600 breeding sows, was reported ASF positive to the veterinary authority by the laboratory on 31st of July 2018. The farm is located in south-western part of Latvia, also in a restriction area due to ASF in wild boar (Figure 1). Pigs were kept indoors in two separate locations (B1 and B2) three km apart. In B1 breeding pigs, including boars, pregnant and farrowing sows with piglets and weaned pigs were kept in 18 stables. Fattening pigs and gilts were kept in six stables in location B2. There were regular movements of different pig consignments between the two locations on a daily basis. The fatteners were sold to slaughterhouses in Latvia and Lithuania.

### 2.3. Epidemiological Investigations

The epidemiological investigations were conducted according to the European Union legislation [11,24] and were performed by the veterinary authority. The investigations included onsite inspections, interviews with relevant personnel from the farm’s administration, veterinarians and workers directly involved in the daily care of the animals. The documentation regarding animals, personnel, visitors and vehicles entering and leaving the farms were checked and the farm biosecurity status was evaluated. Hypothetical virus introduction routes were identified and checked as previously described [10].

For the estimation of HRP and the reconstruction of the spreading pathways within the farm, the information obtained from the farm records and interviews with farm staff, clinical and post-mortem findings and laboratory results were evaluated. If clinical data were interpreted, an infection cycle from infection until death of ten days was assumed [17,18,22]. For laboratory findings showing positive RT-PCR and immunoperoxidase test (IPT) results after a period of up to 10 days, it was assumed that might have passed from infection until death of the sampled pigs (C. Gallardo, personal communication (European Union reference laboratory (INIA-CISA), Valdeolmos, Madrid, Spain).

In addition, samples for laboratory testing were taken before culling from pigs of different groups and different stables to obtain a better picture of virus spread within the farms (Table 2 and Table 3). The samples were tested for the presence of viral genome (RT-PCR test) and ASF antibodies (Ab-ELISA and IPT) at the Institute of Food Safety, Animal Health and Environment “BIOR” (National Reference Laboratory, NRL) according to the manufactures instructions of the test kit.

### 2.4. Chronology of Events in Farm A

Timeline of the disease event in Farm A is summarized in Figure 2. ASF suspicion was notified by the farmer based on severe clinical signs and increased mortality in stable 2 on the 14th of July. The observed clinical signs were abortions, vomiting, lethargy, high fever and loss of appetite. The disease was officially confirmed in the evening of the same day. Four dead pigs sampled on the 14th July were PCR positive and two of them were also antibody positive (ELISA and IPT). The clinical events started most probably in stable 2 where pregnant sows were kept in groups. On 2nd of July one sow, which previously aborted, died. However, the animal was not sampled and tested for ASF and no suspicion was reported. According to the farm workers, the remaining sows of the same pen were moved to other pens within the same stable. In the following days, several other sows aborted and mortality increased, reaching the peak of 12 dead sows on the 11th of July. A total of 43 sows died between the 2nd and 14th of July, while 25 sows were slaughtered due to severe health problems. Five more sows died in the farrowing stable 3, all coming from stable 2. One of them died on 7th of July and the other four on the 8th and 9th of July, respectively.

Additional samples were taken from pigs in all other stables the day after outbreak confirmation to investigate further the disease situation in each separate unit and to reconstruct the potential spread scenario within the farm. Purposive samples were taken from pigs with any disease signs and randomly to detect 10% prevalence with 95% confidence in each sampling unit (stable) [10,17,25,26,27]. The laboratory results are shown in Table 2.

### 2.5. Chronology of Events in Farm B

The ASF suspicion in Farm B was notified to the veterinary authority by the NRL on the 31st of July. Three pigs older than four months died on the 29th of July and tested PCR positive for ASF during routine testing in the frame of enhanced passive surveillance. The pigs were from location B2, stable 4. The timeline of the disease event in that stable is presented in Figure 3. Alongside RT-PCR positive results, the three pigs mentioned above were also positive for ASF antibodies (IPT test). According to the information provided by the owner and the farm veterinarian, no clinical signs indicating ASF were noticed prior to the deaths of the pigs. The ASF positive pigs belonged to a group of 280 weaners, which were moved on the 23rd of July from location B1 stable 7 to location B2 stable 4. A few days after the movement an increased mortality was observed in stable 4 in the group of the moved animals (Figure 3B). The increase in mortality was associated with very hot weather conditions and previous experience with edema disease (*E. coli* enterotoxaemia). However, mortality in location B1 stable 7 did not increase notably (2 to 3 weaned pigs per week). In the farm’s records a slight increase of mortality could be seen in stable 4 of location B1 starting mid-July (Figure 3A).

After disease notification, on the 1st of August samples were taken from pigs of both locations to identify possible virus spread within the farm. Considering the large number of stables and animals, the sampling was targeted to dead pigs and pigs with clinical signs.

In location B1, seven dead pigs from stable 4 were sampled and six tested RT-PCR positive. Four of these pigs tested negative for antibodies to ASF by IPT. In addition, 40 live pigs from different stables were sampled and all of them tested negative for antibodies (ELISA) and virus genome.

In location B2, samples were taken from 13 dead and 51 sick pigs suspected of being infected. These pigs were selected since they were less active and reluctant to move. One dead pig in stable 1 and two dead and five suspect live pigs from stable 4 were found RT-PCR positive. Two RT-PCR positive dead pigs from the stable 4 were positive by IPT. All the suspect live pigs were tested for virus genome and antibodies (ELISA). The five PCR positive suspected pigs were tested for antibodies with both—ELISA test and IPT and three tested positive in IPT. Details of sample size and laboratory results are shown in Table 3.

### 2.6. Estimation of the High Risk Period in Farm A

The first pig, which aborted and died on 2nd July was presumed to be the first animal which got infected (index case) in farm A. Thus, ASF virus could have entered the farm about 10 days earlier, around 22nd of June, considering the maximum survival time of 10 days after the infection with an ASF genotype II virus strain [22]. The suspicion of the outbreak was notified on 14th of July, which makes a total of 23 days of HRP for the farm A. In addition, two dead pigs sampled on 14th July were seropositive (ELISA and IPT). This finding supports the estimate of HRP being longer than 10 days. A further indicator for prolonged HRP was the increased mortality during the last three weeks before notification supporting the hypothesis that more than two infectious cycles should have passed.

### 2.7. Estimation of the High Risk Period in Farm B

Due to the enhanced passive surveillance, conducted on Farm B we assumed that the RT-PCR positive animals which died on the 29th of July were also the first infected animals (index cases) on the farm. In addition to the RT-PCR positive results, these pigs were tested positive also by IPT method. Therefore, we assume that they got infected 7 to 10 days earlier, around the 18th of July. The estimated HRP would be around two weeks.

### 2.8. Potential Spreading Scenario within Farm A

The potential spreading Scenario within Farm A is shown in Figure 4. Most probably, ASF infection started in stable 2 with a sow which aborted and died on the 2nd of July (index case). This sow came from the insemination stable 1 around 10–12 days before. However, the clinical status of the animals in the insemination stable as well as the laboratory results were not indicating an ongoing infection there. Pigs were moved from stable 1 to stable 2 using an outdoor path. It can be hypothesized that the index sow picked up infection from potentially contaminated environment during the outdoor walk. The path which connected two stables was badly constructed, with no solid floor and with partly damaged fencing, which may allow the inadvertent escape of pigs into the farm’s territory. If the farm’s territory was contaminated with ASF virus it was possible for pigs to get infected. After the index sow died, the sows from that pen were moved to other places within the same stable and they were the next ones who aborted and died.

Based on the disease timeline, we assume that the five ASF positive sows in the farrowing stable 3 became infected while still being in stable 2. Most probably, these animals were moved to the farrowing stable during the incubation period.

One weaned pig was found PCR positive in stable 4. The weaners in stable 4 were brought from stable 3. Therefore, it can be assumed that infection in stable 4 was introduced through weaned piglets. ASF virus spread between the stables in Farm A could occur due to movements of infected animals. However, pigs in other stables, where no animal movements took place during the HRP, remained unaffected.

### 2.9. Potential Spreading Scenario in Farm B

The potential spreading scenario in Farm B is shown in Figure 5. The three pigs (index cases) in stable 4 of location B2 tested RT-PCR positive, were moved from location B1, stable 7 six days before they died. Since the pigs were also seropositive, it can be excluded that infection happened during the transport.

The positive RT-PCR results from pigs sampled after disease notification proved that ASF was also present in stable 4 of location B1 and in stable 1 of location B2. Most probably infection started in stable 4, location B1 and then probably jumped over via stable 7 to location B2. This can be justified by the movement records between both locations. However, it remains unclear how the infection spread between stables in location B1. Most probably, it might have happened via movement of farm workers or via shared stable equipment. All stables were connected through corridors where workers and equipment, including wheel borrows, could move without additional biosecurity measures. Furthermore, it remained unclear how the virus entered stable 1 in location B2. Based on the farm’s records, movement of pigs between stables did not take place. Most probably shared equipment or farm workers could have played a role in virus transmission.

## 3. Discussion

### 3.1. Early Detection of the Disease and HRP

Early detection would mean the detection of the very first ASF infected pigs on the farm at the shortest period of time after the infection was established. Ideally, these should be the first pigs which show clinical signs and die due to ASF. This might happen about one week after infection. However, seldom reliable data and information are found indicating the start of an infection (index case) on a farm. Due to the rather low infectivity of the circulating ASF virus, only few animals may become infected and die during the first weeks [19]. Therefore, the general mortality rate may not exceed normal levels and the initially low ASF induced mortality can easily be overlooked, particularly in larger farms. It could be shown that ASF virus might circulate in a herd for nearly a month before it causes a marked increase in mortality [15,17,18]. Under such conditions, enhanced surveillance targeting sick and dead animals, independently from the mortality rate, would be the only practical option for early ASF detection. In contrast, sampling of clinically healthy pigs would fail to detect ASF on a farm during the initial stage.

In Farm A, ASF was suspected and reported due to increased morbidity and mortality of pregnant sows at a late stage. No enhanced passive surveillance system for early detection was in place targeting dead or suspect animals. An HRP of longer than 3 weeks was estimated. In Farm B, the enhanced passive surveillance was implemented and ASF could be detected before mortality noticeably raised. The HRP was around two weeks.

A reasonably accurate estimate of the HRP may allow the veterinary authorities to trace efficiently potential secondary outbreaks or virus spreading via animal movement, contaminated products, vehicles, people and other means. However, under field conditions estimating the HRP is rather difficult and often unsatisfactory due to insufficient or questionable data. The history based on clinical signs and morbidity records might not always be reliable or relevant laboratory examinations might be missing. However, mortality records and laboratory test results proved to be useful indicators for estimating the HRP. Experimental studies with genotype 2 ASF virus showed that in average an infection cycle, the time from infection until death of an animal, is around 10 days [15,16,21,22]. Usually antibodies detectable in an ELISA test are found in animals, which do not die within the first 10 days of infection [22]. The timeline showing the increased mortality on Farm A is a strong indicator that more than two infectious cycles of about 10 days each might have occurred, meaning that more than 20 days must have past, since virus introduction. However, in a worst-case scenario the HRP could have been even longer. Seropositivity detected by ELISA test is a very clear indicator that the virus must have been on the farm for more than ten days. If the concept of weekly testing of dead pigs would have been in place on Farm A, the outbreak could have been detected much earlier, most probably when the first sow was found dead. On Farm B none of the pigs tested positive by ELISA but the first PCR positive animals were also positive in IPT. Such results indicate a relative recent infection. Seroconversion can be seen by IPT around 7 to 10 days after infection [28], (C. Gallardo, personal communication (European Union reference laboratory (INIA-CISA), Valdeolmos, Madrid. Spain).

The sooner the early detection surveillance scheme for detecting potentially infected holdings is implemented, the earlier farms are found in which only one or few virus positive animals are present. This leads to the dilemma regarding the acceptance of the depopulation measures, which have to follow without any delay. Due to this dilemma, alternative culling and surveillance schemes for large farms where only few infected animals are present could be developed.

### 3.2. Virus Spreading within the Farms

Reconstructing the potential spreading pathway of the virus within the farm may help to identify how and where the virus might have entered and to which degree the farm was affected. Additional blood or organ samples for laboratory examinations taken from suspect animals, from all production units during culling or shortly before, can prove the presence or absence of the virus in different farm units. However, under field conditions, sampling of pigs before culling or during culling is difficult because of time pressure and other logistical arrangements. In Farm A the ASF infection most probably started in the stable 2 and spread to the stables 3 and 4, while four other stables were not affected. For Farm B reconstruction of the virus pathway was more difficult compared to Farm A, mainly due to the complex epidemiological situation ongoing in parallel in the two separate locations. The clinical events started most probably in stable 4, location B1 and then jumped over via stable 7 to location B2. Direct contacts with infected live pigs or carcasses posed the highest risk of spreading the infection [9,29]. As in the case of Farm A, we could see how infection spread from one stable to the next following the movements of infected pigs. Non-compliance with farm biosecurity measures left space for virus introduction and spread within the farm (biosecurity [30].

Experimental findings as well as field studies have proven a low transmissibility of ASFV, which may delay the virus spread inside a pen and between pens [15,16,18]. The calculated period from infection until onset of infectiousness was around 10 days within a pen and around 13 days between pigs in different pens in a study by Guinat et al. [15]. Our findings support these experimental studies, confirming the relatively slow spread of the infection and are based on the analysis of ASF spread within both farms, considering the results of clinical and laboratory investigations, as well as on the estimated length of the HRP. Although relatively long HRP was detected, particularly for Farm A, there were still negative animals within the affected stables and there were stables not affected at all on the date of investigation. However, the speed of virus spread is very much influenced by the local biosecurity measures, the local farm infrastructure and the farm management. Therefore, daily maintenance of high biosecurity measures must have highest priority [9,31]. Farms located in infected areas due to ASF virus circulation in wild boar populations are under particular risk.

## 4. Conclusions

ASF rather spreads slowly within pig herds, particularly in the initial stage of the outbreak. Several weeks can pass with ASF infection within the farm until mortality rises above the average level.

Enhanced passive surveillance by testing dead or sick animals is an effective method for the early detection of ASF in large pig farms. In addition, it may lead to a considerable reduction of the HRP.

Mortality records and laboratory results proved to be useful indicators for estimating the HRP.

With early detection and a short HRP it can be assumed that large parts of the farm are not yet affected by ASF. Under the precondition of slow virus spread, non-affected farm units could be excluded from culling measures. Good farm management and strict internal biosecurity measures as well as an intelligent surveillance scheme should be in place to reach this goal.

Veterinary authorities may take into consideration slow spread of the virus within a farm when planning and performing stamping out procedures thus gaining time for epidemiological investigations, sampling for laboratory investigations, as well as time to prepare properly for culling and synchronize it with carcass destruction.

## Figures and Tables

**Figure 1 vetsci-07-00105-f001:**
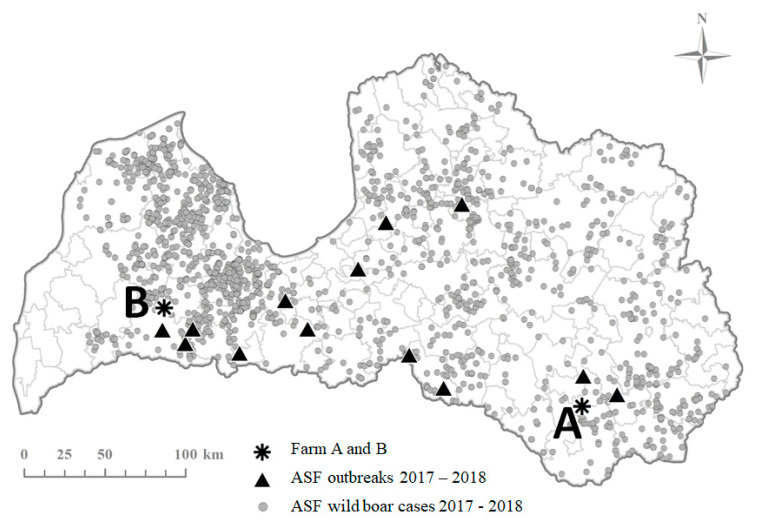
ASF in Latvia in 2017–2018 and the location of the two outbreak farms described in this study.

**Figure 2 vetsci-07-00105-f002:**
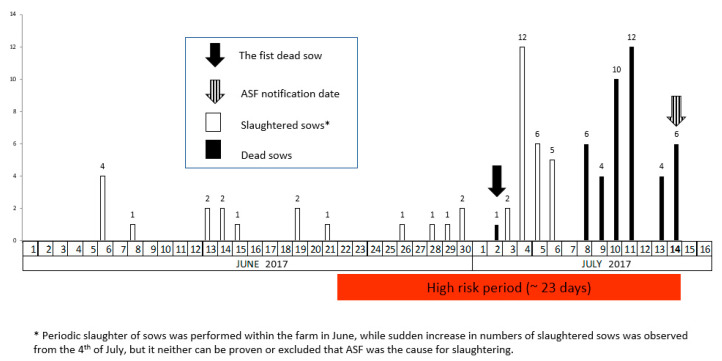
Timeline of the disease event in Farm A, stable 2 and the estimated high risk period.

**Figure 3 vetsci-07-00105-f003:**
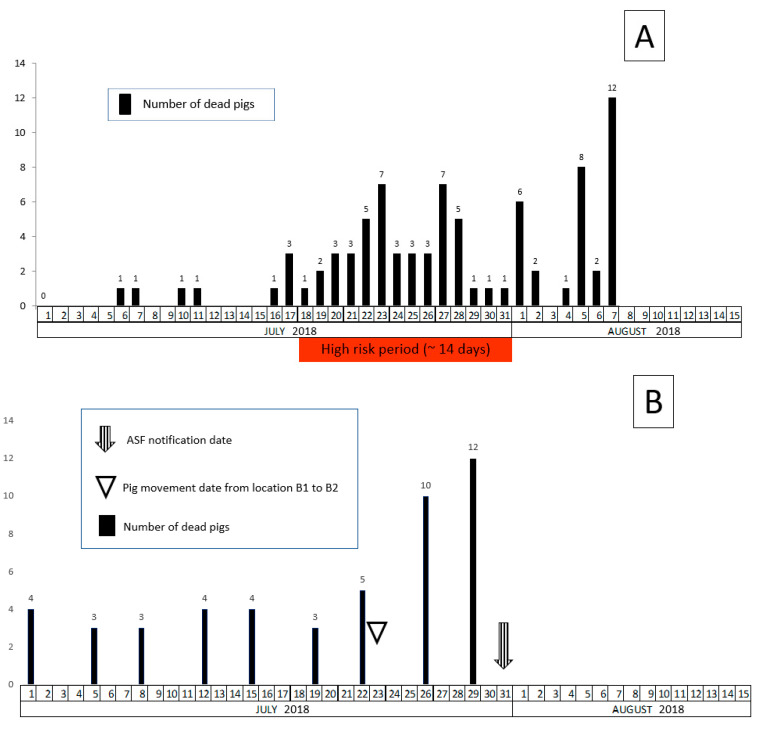
Timeline of the disease event in location B2 stable 4 (**A**) and B1 stable 4 (**B**) and the estimated high risk period for farm B.

**Figure 4 vetsci-07-00105-f004:**
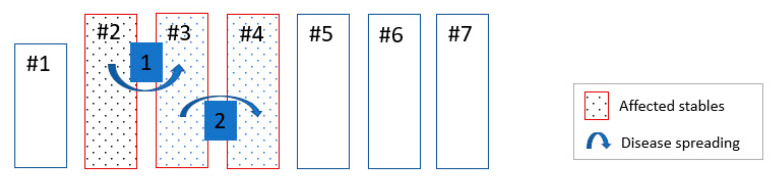
Potential spread of ASF virus between the stables in a Farm A.

**Figure 5 vetsci-07-00105-f005:**
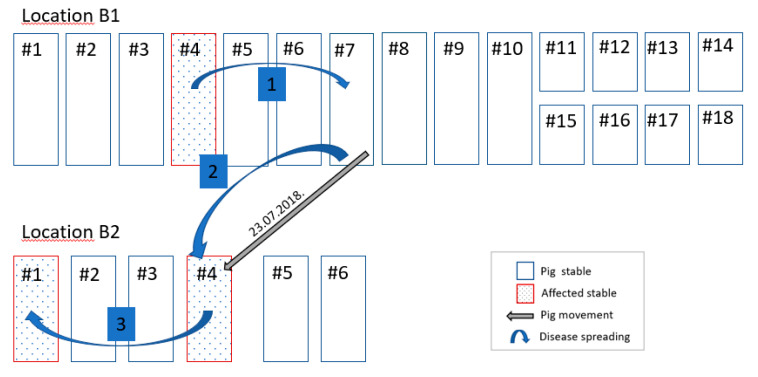
Potential spreading of ASF virus between the stables in a Farm B.

**Table 1 vetsci-07-00105-t001:** African swine fever (ASF) outbreaks and cases in Latvia during 2014–2020.

Year	Outbreaks in Domestic Pigs			Cases ** in Wild Boar	Territory Affected by ASF in Wild Boar (%) ***
	Commercial *	Backyard *	No of Culled Pigs		
2014	13	19	552	247	31.83
2015	4	6	213	1048	51.03
2016	3	0	308	1146	78.08
2017	5	3	19,914	1431	87.47
2018	5	5	16,705	905	95.01
2019	1	0	52	430	96.09
2020 (1st June)	0	0	0	133	96.39
Total	31	33	37,744	5340	N/A

* pig farms are categorized according to the Strategic approach to the management of African Swine Fever for the EU, SANTE/7113/2015—Rev 12. ** Case is defined according to the European Union (EU) legislation [11]. *** Estimate based on the size of restriction zones by the end of a particular year. Restrictions zones are defined according to the EU legislation [12,13].

**Table 2 vetsci-07-00105-t002:** RT PCR and serology results of pigs of Farm A sampled in different stables before culling.

Stable Id	Number of Pigs in the Stable	Category of Pigs	Number of Samples Taken in a Stable	Laboratory Testing Results	
				RT PCR Positive	Serology (ELISA, IPT) Positive
1	213	Boars, sows	3	0	0
2	301	Sows *	4	4	2
3	696	Sows **	28	10	0
4	1761	Weaners	28	1	0
5	1157	Fatteners	29	0	0
6	1136	Fatteners	28	0	0
7	554	Fatteners	20	0	0
Total	5818	N/A	140	15	2

* Sows kept in group pens. ** Sows kept with piglets.

**Table 3 vetsci-07-00105-t003:** RT PCR and serology results of pigs sampled before culling in different stables of Farm B at location B2.

Stable Id	Number of Pigs in the Stable	Category of Pigs	Number of Samples Taken in a Stable	Laboratory Testing Results	
				RT PCR Positive	Serology (IPT) Positive
1	685	Gilts, Fatteners	13	1	0
2	877	Gilts, Fatteners	10	0	0
3	946	Gilts, Fatteners	12	0	0
4	997	Gilts, Fatteners	18	7	5
5	848	Gilts, Fatteners	4	0	0
6	1118	Fatteners	7	0	0
Total	5471	N/A	64	8	5

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
