# Peer review of "African Swine Fever in Two Large Commercial Pig Farms in LATVIA—Estimation of the High Risk Period and Virus Spread within the Farm"

_vetsci, 2020, doi:10.3390/vetsci7030105_

Round 1

Reviewer 1 Report

The authors presented a case report of ASF from the context of two farms to evaluate transmission potentials of ASFv.

The work was well designed and well implemented. Suggestion for improving the manuscript have been inserted into the returned document. 

The author should adhere to implementing the changes and addressing the queries.

Author Response

We would like to thank for the careful review, all comments and suggestions made to improve the manuscript.

We have followed all the recommendations inserted in the manuscript by the reviewer in the text and we have corrected accordingly the text. We thank for these valuable recommendations; they really improve the meaning of our manuscript.

Comment 1: Concerning the suggestion in line 74 about the additional figure to demonstrate spatial displacement of the stables, we would like to clarify that the layout of the stables are demonstrated in Fig.4 and Fig.5 in the context of the potential virus spreading pathways between stables. Precise number of stables are shown in both figures (4 and 5).

Comment 2: Recommendations in line 78 are addressed by inserting the veterinary authority to whom the suspicion in case of Farm 1 was notified (lines 82-83) and about laboratory confirmation notification in case of Farm B (lines 93-94). In addition, the date of notifications are added (lines 82 and 94). The first observations of clinical signs in pigs in both farms are described in sections 2.4 “Chronology of events in Farm A” and 2.5. “Chronology of events in Farm B”.

Comment 3: Comment in the line 105 is addressed by inserting the references to the publications were the interpretation of 10 days as a maximum survival time of ASF infected pigs are described (line 111). To respect the same principle the reference is inserted also for interpretation of laboratory results (lines 113-114).

Comment 4: Suggestions in line 131 are addressed accordingly. The timeline was revised by inserting the year and additional footnote explaining the periodic slaughter of sows in the farm (line 139).

Comment in the line 193 is addressed by adding explanation on how the escape of pigs from the outdoor path could pose the risk of getting infection (lines 202-203). Hypothesis of contacts with wild boar we have excluded, since all farm was completely fenced.

Comment in line 237 is addressed by inserting the explanation that we are talking about the pregnant sows (line 248).

Line 282 –290: the spacing is fixed. Thank you very much for all the efforts made!

P.S. More reference documents are added to the manuscript and therefore all references numbers and list of reference documents is amended.

Reviewer 2 Report

This manuscript, ID vetsci-875188, by Lamberga et al., is a case report presenting an estimation of the high risk period and virus spread within two pig farms affected by African swine fever virus. In my opinion, the information presented is of interest for those working in the field and I only have a few minor comments.

- A brief definition of “case” and “restriction zone” would contribute for clarity

- Line 55, “the slow spread of the disease within a farm” is a central aspect in this work and, therefore, I suggest a more detailed presentation and a wider bibliographic support for within farm transmission parameters.

- In the description of events in farm A, it reads that the suspicion was based on “severe clinical signs” (line 115) and that 25 sows were slaughtered “due to severe health problems” (line 123). If available, a description of these clinical signs / health problems should be included, particularly because in farm B “no clinical signs indicating ASF were noticed” line 142).

- Line 173, “average survival time of 10 days …” should read “maximum survival time of 10 days …”. In the reference cited, reference 13, Blome et al. 2013, it reads that “mortality was 100% in less than 10 days”.

- Lines 263-265, this sentence should be better explained, perhaps changing the words paradox / dilemma.

- Line 292, in the section “conclusions and recommendations”, the claim “ASF spreads slowly within pig herds” is too strong. This cannot be concluded from the observations presented and, as stated a few lines above (287-288), “the speed of virus spread is very much influenced by the local biosecurity measures, the local farm infrastructure and the farm management”.

Author Response

We would like to thank the reviewer for a positive response and valuable improvements suggested.

Comment 1: A brief definition of “case” and “restriction zone” would contribute for clarity

Replay to comment 1:  Thank you for picking up this point. There are new footnotes inserted in Table 1 lines 48-50 to clarify the definitions of “case” and “restriction zone”.

Comment 2: Line 55, “the slow spread of the disease within a farm” is a central aspect in this work and, therefore, I suggest a more detailed presentation and a wider bibliographic support for within farm transmission parameters.

Replay to comment 2: Both recommendations are addresses, in the line 58 two more reference documents are inserted where slow spread of ASF virus is mentioned (reference 10 and 19). We have added an explanation about slow spread of ASF virus in lines 297-301.

Comment 3: In the description of events in farm A, it reads that the suspicion was based on “severe clinical signs” (line 115) and that 25 sows were slaughtered “due to severe health problems” (line 123). If available, a description of these clinical signs / health problems should be included, particularly because in farm B “no clinical signs indicating ASF were noticed” line 142).

Replay to comment 3: We have amended the manuscript as suggested and inserted the observed clinical signs in Farm A (line 123).

Comment 4: Line 173, “average survival time of 10 days …” should read “maximum survival time of 10 days …”. In the reference cited, reference 13, Blome et al. 2013, it reads that “mortality was 100% in less than 10 days”.

Replay to comment 4: the suggestion is addressed and word “maximum” is inserted instead of “average” in line 181

Comment 5: Lines 263-265, this sentence should be better explained, perhaps changing the words paradox / dilemma

Replay to comment 5: Indeed, the suggested word “dilemma” expresses more precisely the meaning and the text is changed accordingly, line 274.

Comment 6: Line 292, in the section “conclusions and recommendations”, the claim “ASF spreads slowly within pig herds” is too strong. This cannot be concluded from the observations presented and, as stated a few lines above (287-288), “the speed of virus spread is very much influenced by the local biosecurity measures, the local farm infrastructure and the farm management”.

Replay to comment 6: Thank you, we have addressed the suggestion and inserted the word “rather” in line 306. This recommendation is also related with comment No 2 and we do consider that the amendments made regarding slow spread of the virus are no better addressed.

Thank you very much!

P.S. More reference documents are added to the manuscript and therefore all references numbers and list of reference documents is amended.